# Migrant-friendly maternity care in Montreal, Canada: A cross-sectional study on migrant women's care perspectives

Isabel Baltzan[1], Lisa Merry[2]*, William Fraser[3], Sonia Semenic[1], Sandra Pelaez[4], Alexis Edington[1], Ayesha Baig[1], Anita Gagnon[1]

1 Ingram School of Nursing, McGill University, Montreal, Quebec, Canada, 2 Faculté des sciences infirmières, Université de Montréal, Montreal, Quebec, Canada, 3 Centre hospitalier universitaire de Sherbrooke, Université de Sherbrooke, Sherbrooke, Quebec, Canada, 4 École de kinésiologie et des sciences de l'activité physique (EKSAP), Faculté de Médecine, Université de Montréal, Montreal, Quebec, Canada

* lisa.merry@umontreal.ca

## Abstract

### Objective

We assessed the extent to which recommended migrant-friendly maternity care (MFMC) components were provided to recently-arrived international migrants giving birth in Montreal, Canada, and the extent to which the provision of MFMC components was related to socioeconomic and migratory characteristics.

### Methods

We conducted a cross-sectional study of migrant women giving birth in four hospitals in 2014–2015. Data were collected using the Migrant-Friendly Maternity Care Questionnaire (MFMCQ), focusing on access to prenatal care, communication facilitation, healthcare provider (HCP) support, and responsiveness to preferences for care. Data were analyzed descriptively and through logistic regression.

### Results

Of 2636 participants, most reported always being kept informed (86.1%) and finding HCPs helpful (90.3%), although 22.9% reported barriers to accessing services during pregnancy, and only 11% or less were asked about care preferences. Of 847 needing interpreters, 84.7% reported not being offered any. Worse access to prenatal care was reported among women who had arrived more recently [OR 0.55, 95% CI 0.36, 0.85], had lower income [0.69 (0.52, 0.90)], or had less education [0.66 (0.47, 0.94)]. Low language ability was most often associated with inadequate MFMC [e.g., worse HCP support during pregnancy [0.56 (0.36, 0.87)] and worse responsiveness to preferences for care during labour [0.55 (0.31, 0.98)]]. Maternal region of birth was associated both positively and negatively with all MFMC components.

**Data availability statement:** Data are not publicly available due to the sensitive nature of information gathered and because participants did not provide consent for the data to be shared. For researchers who are interested in knowing more about the data, or who may seek to use it for further research, they may communicate with the Institutional Review Board for the Faculty of Medicine and Health Sciences at McGill University, via email at submit2irb.med@mcgill.ca.

**Funding:** Funding for this research was provided by the Canadian Institutes of Health Research (CIHR).

**Competing interests:** The authors have declared that no competing interests exist.

## Conclusion

Although some MFMC has been implemented, gaps remain. Addressing language barriers remains a top priority. To deliver optimal MFMC, HCPs and policymakers should provide care that is responsive to women's socioeconomic and migratory backgrounds.

## 1. Introduction

The health and care of international migrants is strongly influenced by mechanisms beyond conventional biomedical risk factors, such as restricted access to care, language barriers, and differing quality of care. [1–6] Migrants' health outcomes also vary between migrant categories, with undocumented migrants, asylum seekers, and refugees presenting worse health outcomes and access to care than economic and family-sponsored migrants, for example. [1,7–10] As international migration continues to increase each year, ensuring that migrants have equitable access to healthcare that addresses their needs becomes more critical. [11] In particular, maternal and perinatal health discrepancies between migrant women and their receiving-country counterparts have been described by international research initiatives such as the Reproductive Outcomes And Migration (ROAM) collaboration. [12,13] ROAM's studies have found that some migrant women are at greater risk of adverse outcomes such as preterm birth, emergency caesaerean birth, and perinatal or infant mortality. [14–20] Some migrant women report receiving poorer care, having more health problems post-birth, and having these problems addressed less, both immediately and months later. [7,8,21,22] Throughout Canada, care is dispensed through a public healthcare system, where hospitalization and physician care (at minimum) are paid by provinces for citizens and permanent residents. [23] Access to public healthcare insurance depends on immigration class and is subject to administrative delays. Recent humanitarian migrants may qualify for the Interim Federal Health Program until they are entered into the provincial medicare system; other classes of migrants require private health insurance until they have access to the public program. [24]

Barriers to adequate maternity care include difficulty accessing care during pregnancy and after birth, communication barriers, and substandard care. [7,19,21,22,25–27] Difficulties in access to care can stem from regulatory restrictions that limit migrants' eligibility for healthcare financing in receiving countries, [4,28] and from the challenge of navigating a foreign healthcare system, among others. [1,7,19,21] Specifically, being a new migrant, primiparous, or never having given birth in a new country before, are associated with more challenging navigation of the healthcare system. [1,19,21,29,30] Maternal region of birth and migration status are also associated with having different expectations and experiences of new healthcare systems. [1,29–32] Barriers to communication can negatively affect quality of care, user satisfaction, use of diagnostic testing, use of preventive services, use of unnecessary invasive procedures, continuity of care, and harmful

health outcomes including mortality. [1,2,5–7,19,27,33,34] Inequitable health care has also been described, such as negative care experiences, racism and discrimination. These barriers in turn influence continuity of care and trust in the healthcare system. [3,7] Disparate care access and delivery appears to affect younger, low-income, and less educated women more than others. [7,25,29,32]

Recommendations have been made to healthcare professionals (HCPs) to promote migrants' general and perinatal health and health literacy. [35,36] These recommendations, such as communication facilitation, promotion of social support, education to reduce high body mass index (BMI), treatment of pre-pregnancy and maternal illnesses, early access to prenatal care, and responsiveness to preferences for care, are key to providing migrant-friendly maternity care (MFMC). [35]. Though maternity care is a determinant of international migrant women's health, the extent to which MFMC components are provided remains unclear. The influence of migrant women's socioeconomic and migratory backgrounds on specific care components also remains understudied, as do migrant's perspectives of care. Further, delivery of MFMC has become more relevant due to extensive global migration and thus, warrants better understanding.

Using existing recommendations for MFMC and converting these into measurable components, ROAM developed the Migrant-Friendly Maternity Care Questionnaire (MFMCQ). This tool captures migrant women's perspectives regarding the application of care recommendations during pregnancy, childbirth and postpartum; it also includes items to gather sociodemographic and migration background information. [37] The MFMCQ was developed by identifying MFMC-related questions from existing questionnaires, validating them through a Delphi consensus process, translating and back-translating them, and having them culturally validated by recent migrant mothers from a range of countries (see Translation Protocol in S1 Appendix). [37,38] The eight different language versions of the MFMCQ used in the current study can be found in S2-S9 Appendices. The MFMCQ has also been adapted and/or translated for use in other language settings, including Dari, Farsi, German, Norwegian, Persian, Polish, Portuguese, Somali, Swedish, Tigrinya, and Turkish. [32,39–48]

In Montreal, an urban city where ≥50% of all births are to migrant women, we sought to assess to what extent recommended MFMC components were being provided to recently arrived (≤8 years) international migrant women giving birth in one of four urban hospitals. We did so by administering the MFMCQ to a sample of this population. Our aim is to add to the body of knowledge in the field of maternity migrant care by answering the following research questions: (1) *To what extent are certain recommended components of MFMC being provided to recently arrived international migrant women giving birth in an urban Canadian city?* (2) *To what extent is provision of these MFMC components dependent on migrants' socioeconomic characteristics and migration background?*

## 2. Materials and methods

### 2.1. Study design & setting

We conducted a cross-sectional study in 2014–2015 on the post-partum units of four public university-affiliated hospitals in Montreal, selected for their high proportion of births to international migrants (42.3–70.7% of births to immigrant women per hospital in 2011). [49] One hospital is smaller and provides primary and secondary care, while the other three larger hospitals provide tertiary (or quaternary) care. Each hospital has between 217 and 637 beds and serves a diverse population in central areas of the city. The study was granted ethics approval from the McGill Faculty of Medicine Institutional Review Board and by the ethics committees of each participating hospital.

### 2.2. Study population

Of the 11,755 women who gave birth in the four target hospitals over the course of data collection, 2962 women were confirmed to meet the eligibility criteria and 2636 of these women (89%) consented and participated in the study (Fig 1).

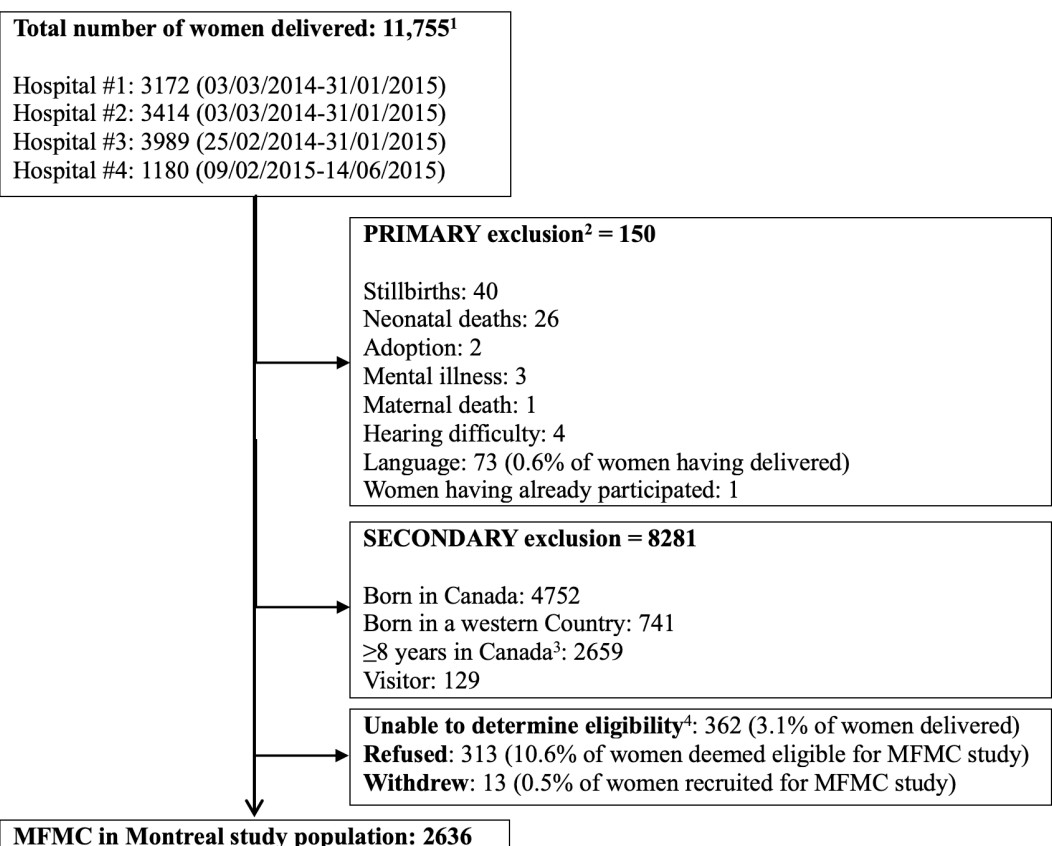

**Fig 1. The "Migrant-Friendly Maternity Care in Montreal" study population, from four hospitals in Montreal, Canada between March 2014 and June 2015.** [1]227 twins and 7 sets of triplets. [2]Primary exclusion criteria determined by hospital birthing log and/or medical chart or if unable to communicate with woman (i.e., exclusion did not require administration of a questionnaire). [3]Initially exclusion was > 5 years. It was changed to ≥8 years after 5 months of recruitment to optimize the recruitment rate. [4]Women were discharged or declined to be approached or research personnel were restricted by hospital staff to approach women.

Participants were included if they had lived in Canada for 8 years or less and spoke at least one of the eight study languages (Arabic, English, French, Hindi, Chinese (Mandarin/Cantonese), Spanish, Urdu, or Vietnamese). Participants could withdraw from participating at any time, and were informed that no information would be shared with immigration authorities. Exclusion criteria were: major hearing impairments; major mental illness or cognitive impairment precluding fully informed consent; "Western" country/region of birth (i.e., USA, Western and Northern Europe including Scandinavia, Israel, Australia, Japan); and stillborn/neonatal deaths (to avoid burdening the mother).

## 2.3. Data collection

Multi-lingual research assistants explained the study to eligible participants in postpartum units, with the help of interpreters if needed or requested, and obtained consent. Thereafter, they interview-administered the MFMCQ to all participants in one of eight languages (S2-S9 Appendices). The MFMCQ was used to collect information on experiences during pregnancy, labour, and post-birth, as well as participants' socio-demographic characteristics and migration background. In addition, research assistants administered a short supplementary questionnaire (also available in the eight study languages as S10-S17 Appendices) and recorded data from medical records on a standardized form for each participant (S18 Appendix).

## 2.4. Analyses

Descriptive analyses were carried out on socio-demographic characteristics, migration indicators, and selected MFMC components: access to prenatal care, communication, and experience and perceptions of care (including HCP support and responsiveness to preferences for care). Other MFMC components: 'education to reduce high BMI' and 'identification and treatment of pre-pregnancy/pregnancy illnesses' were examined but were not analyzed further due to poor data reliability as a result of poor recall and inaccurate self-reporting. Some MFMC questions were specific to pregnancy, labour, and post-birth phases, while others assessed women's overall perceptions of maternity care. Results were presented for each phase when they differed across phases by at least 10%. When responses differed by <10%, results were combined and presented for all phases together. For questions with the option to write-in an explanation under 'other,' written responses were sorted by theme to minimize data loss.

To address the second research question, we used logistic regression modelling. We estimated associations between eight potential explanatory variables and seven MFMC components. Regression models focused on pregnancy and labour; we did not include post-partum outcomes in regression modelling since the participants were still in hospital when data were collected, such that the data do not reflect their complete post-partum experiences. Four MFMC components were measured during pregnancy and three during labour. These were operationalized as follows:

(1) **Early access to prenatal care**, defined as having care provided (MFMCQ#4), having care begin at ≤3 months (MFMCQ#6) and having ≥4 prenatal care visits (MFMCQ#7);

(2) **Communication facilitation during pregnancy,** defined as having information given *in their language* during pregnancy (MFMCQ#13) and *always* having someone to interpret (or N/A) (MFMCQ#58a) and *always* understanding information given (MFMCQ#55a);

(3) **Communication facilitation during labour,** defined as *always* having someone to interpret (or N/A) (MFMCQ#58b) and *always* understanding information given (MFMCQ#55b);

(4) **Healthcare provider (HCP) support during pregnancy,** defined as *always* finding HCPs reassuring and/or encouraging (MFMCQ#74a);

(5) **Healthcare provider (HCP) support during labour,** defined as having a choice of support during labour (or N/A) (MFMCQ#29), having a companion during labour (MFMCQ#30), and *always* finding HCPs reassuring and/or encouraging (MFMCQ#74b);

(6) **Responsiveness to preferences for care during pregnancy,** defined as having been asked about preferences for care (MFMCQ#16) and *never* having decisions made without wishes considered (MFMCQ#73a);

(7) **Responsiveness to preferences for care during labour,** defined as having been asked about preferences for care (or N/A) (MFMCQ#32) and *never* having decisions made without wishes considered (MFMCQ#73b).

The eight explanatory variables included socio-demographic and migration variables known to be important for migrant health [8,20,22] and with ≥10% variation in responses: previous birth in Canada (Y/N), length of time in Canada (<2 years, 2–5 years, or >5 years), parity (Multiparous/Primiparous), language ability in English and/or French (speaking with difficulty or not at all in both; speaking fluently or well in one and with difficulty or not at all in the other; or speaking fluently or well in both), household income (≤CAD21,000, or>CAD21,000 in 2014), level of education (≤secondary completed, or>secondary completed), maternal age in years (≤35, or >35), and maternal region of birth (per World Bank analytical groupings: Sub-Saharan Africa, Middle-East/North Africa, East/South-East Asia, South Asia, Eastern Europe, or South America/Caribbean). [50] Reference categories for each variable were the group with the best expected health outcomes based on the literature, except for maternal region of birth. For the latter, *not* being from the region in question was the reference category (e.g., *not* being from Sub-Saharan Africa for the variable Sub-Saharan Africa); separate models were

generated to test for each region. Participants who had responded to all explanatory variable questions and had complete outcome data were included in the models. We excluded those missing information to avoid making assumptions about missing data; sample loss was moderate ranging from 16–18%. Confidence intervals were set at 95%. Once the full seven models were completed, we reduced each one, removing explanatory variables with the highest p-values, one at a time; variables were retained if a change in any of the ORs exceeded 10%. We ensured the reduced models included the same populations as in the full models. Care outcomes were coded such that ORs > 1 indicated a beneficial association between the explanatory variable and care outcome, and ORs < 1 indicated an adverse association. IBM SPSS Version 27 was used for statistical analyses.

## 3. Results

### 3.1. Background

Sociodemographic characteristics are reported in Table 1. Nearly all women were married or in a civil union (97.3%). Just over a quarter were 35 years or older (27.5%). The sample of women was highly educated, with virtually all having some level of education (99.4%), and two thirds (66.9%) having completed post-secondary education. Most of the women had an annual household income above CAD21,000 (68.4%); those whose income fell below that were considered lower income, per Statistics Canada's classification for low household income (CAD21,359 in 2014). [51]

Migration descriptors are reported in Table 2. Nearly half of the participants were from the Middle-East/North Africa (43.6%), though all world (non-Western) regions were represented in the sample. Just over half of the women had been in Canada 2–5 years (54.1%). Two thirds of the women (67.1%) had never given birth in Canada before, with just under half being primiparous (44%), and most not having been pregnant upon arrival to Canada (91.5%). Most of the women were either family-sponsored migrants (50%) or fell within the economic migrant/temporary resident/diplomat/student category (45.7%); 4.3% of participants were humanitarian migrants. The majority of participants did not use English or French as the primary language spoken at home (71.1%), although most spoke one or both languages well or fluently (90.1%).

### 3.2. Extent of MFMC components provided

MFMC components related to access to care during pregnancy are reported in Table 3. Most women had access to care from an obstetrician-gynecologist (OB/GYN) in Canada (87.1%), and a majority had timely access to prenatal care (88.7%). Nearly one third (31.4%) of women did not receive one or more services during pregnancy that they would have wanted to have; the services most often wanted but not received were pregnancy/childbirth classes (60.3%) and food banks (22.5%). Most women did not report barriers to accessing care or services (77.1%). For women who reported experiencing barriers, the most commonly reported ones related to knowing whether and where services were offered, and whether they were eligible to receive them (65.7%). Not having time, having to work, or forgetting to seek or attend an appointment were also frequently reported as reasons for not receiving care or services (39.1%).

MFMC components specific to communication during all phases of maternity care are reported in Table 4. Just over one third of mothers felt that they had received enough information on all topics during pregnancy (34.8%), and one quarter felt that they were underinformed on more than three topics (24.7%). Most women used the internet and cellphone applications as a source of information during pregnancy (61.5%). Two thirds of women reported that HCPs gave them information about breastfeeding (69.6%) and helped them with breastfeeding immediately after birth (63.0%). Most women always felt HCPs kept them informed (86.1%), and always understood the information HCPs provided throughout their maternity care (86.9%), though over a quarter would have understood the information better in another language (28.7%). Among women who needed interpreters, over half always had someone present to interpret (52.3%), while one fifth never had anyone present to interpret (21.7%). Professional interpreters were rarely offered (15.3%), although satisfaction was

**Table 1. Socio-demographic characteristics.**

| Socio-demographic characteristics | n (%) |
|---|---|
| **Maternal age (n=2636)** | |
| Up to 26 years | 268 (10.2) |
| 26-34 years | 1644 (62.4) |
| 35 years or older | 724 (27.5) |
| **Parity (n=2636)** | |
| Primiparous | 1159 (44.0) |
| Multiparous | 1477 (56.0) |
| **Highest level of education completed (n=2634)** | |
| None or primary | 50 (1.9) |
| Secondary | 335 (12.7) |
| Postsecondary | 1761 (66.9) |
| Graduate studies | 488 (18.5) |
| **Household income (n=2222)** | |
| Below CAD10,999 | 321 (14.4) |
| CAD11,000 to CAD20,999 | 383 (17.2) |
| CAD21 000 to CAD40,999 | 721 (32.4) |
| CAD41 000 to CAD60 999 | 409 (18.4) |
| CAD61,000 to CAD80,999 | 204 (9.2) |
| Above CAD81,000 | 184 (8.3) |
| **Marital status (n=2634)** | |
| Married/civil union | 2563 (97.3) |
| Single | 51 (1.9) |
| Separated/divorced | 20 (0.8) |
| **Household composition (n=2623)** | |
| Partner | 2363 (90.1) |
| Partner and others (e.g., grandparents, friends) | 155 (5.9) |
| Others (e.g., grandparents, friends) | 64 (2.4) |
| Alone | 41 (1.6) |
| **Lives with children other than the newborn (n=2635)** | |
| Yes | 1239 (47.0) |
| No | 1396 (53.0) |
| **Lives with newborn's father (n=2630)** | |
| Yes | 2484 (94.4) |
| No | 146 (5.6) |

high when interpreters were used (94.3%). Family members/friends acted as interpreters at least once for a substantial proportion of women (61.7%).

MFMC components relating to experiences and perceptions of maternity care (all phases) are reported in Table 5. Women largely found HCPs always respectful (94.5%), helpful (90.3%), and welcoming (88.2%). Most (94.6%) never felt discriminated, however when they did, it was most often associated with language (36.9%) or race (26.2%). Most mothers (79.4%) found that HCPs always spent enough time providing explanations, 79.1% reported that HCPs always asked if they had questions, and 57.7% found HCPs were never rushed. Most women never or rarely waited too long for care during labour (88%) or post-partum (86.8%), though during pregnancy, more women reported sometimes or always waiting too long for care (39%).

**Table 2. Migration-related indicators (N = 2636 unless otherwise stated).**

| Migration-related indicators | n (%) |
|---|---|
| **Region of birth** | |
| East Asia or South-East Asia | 367 (13.9) |
| Eastern Europe | 189 (7.2) |
| Middle-East or North Africa | 1148 (43.6) |
| South America | 288 (10.9) |
| South Asia | 215 (8.2) |
| Sub-Saharan Africa or Caribbean | 429 (16.3) |
| **Length of time in Canada** | |
| Less than 2 years | 788 (29.9) |
| 2-5 years | 1425 (54.1) |
| More than 5 years | 423 (16.0) |
| **Arrived in Canada pregnant** | |
| Yes | 224 (8.5) |
| No | 2412 (91.5) |
| **Gave birth in Canada before (n = 2624)** | |
| Yes | 863 (32.9) |
| No | 1761 (67.1) |
| **Immigration class** | |
| Economic immigrant/temporary resident/diplomat (including students) | 1205 (45.7) |
| Family-sponsored | 1317 (50.0) |
| Humanitarian (refugee history, asylum seeker, no status) | 114 (4.3) |
| **Health insurance** | |
| Medicare (public provincial funding) | 2502 (94.9) |
| Private insurance only | 71 (2.7) |
| Interim Federal Health Program for refugee/asylum claimants | 40 (1.5) |
| None | 23 (0.9) |
| **Language spoken most often at home (n = 2616)** | |
| Arabic or Arabic and another language (other than English and French) | 755 (28.9) |
| Chinese or Chinese and another language (other than English and French) | 214 (8.2) |
| English and/or French | 386 (14.8) |
| English and/or French and another language | 370 (14.1) |
| Other languages other than English or French | 692 (26.5) |
| Spanish or Spanish and another language (other than English and French) | 199 (7.6) |
| **English and/or French spoken ability** | |
| High: Fluent in both, or fluent in one and well-spoken in the other | 751 (28.5) |
| Medium: Fluent or well-spoken in one, and not at all or with difficulty in the other | 1624 (61.6) |
| Low: Not at all or with difficulty in both | 261 (9.9) |
| **Language of Migrant-Friendly Maternity Care Questionnaire (MFMCQ)** | |
| Arabic | 229 (8.7) |
| Chinese | 163 (6.2) |
| English | 770 (29.2) |
| French | 1392 (52.8) |
| Spanish | 56 (2.1) |
| Other (Hindi, Urdu, or Vietnamese) | 26 (1.0) |
| **Region of birth of newborn's father (n = 2617)** | |
| Asia or South-East Asia | 331 (12.6) |

*(Continued)*

**Table 2.** (Continued)

| Migration-related indicators | n (%) |
|---|---|
| Eastern Europe | 157 (6.0) |
| Middle-East or North Africa | 1134 (43.3) |
| North America or North and/or Western Europe | 134 (5.1) |
| South America | 242 (9.2) |
| South Asia | 210 (8.0) |
| Sub-Saharan Africa or Caribbean | 409 (15.6) |

**Table 3. MFMC components provided: Access to care during pregnancy.**

| Access to care during pregnancy | n (%) |
|---|---|
| **Timely prenatal care based on arrival in Canada, i.e., began ≤ 3 months pregnant, or shortly after arrival if arrived in Canada pregnant (n = 2620)** | 2324 (88.7) |
| **Adequate number of prenatal visits, i.e., ≥ 4 (n = 2581)** | 2541 (98.5) |
| **Type of healthcare provider (HCP) that provided care during pregnancy (n = 2599)** | |
| Accessed obstetrician-gynecologist (OB/GYN) in Canada (may have had other HCPs too) | 2263 (87.1) |
| Accessed midwife/general practitioner/nurse practitioner in Canada (no OB/GYN accessed) | 329 (12.7) |
| No care in Canada | 7 (0.3) |
| **Number of services wanted during pregnancy but not received (n = 2627)** | |
| Not applicable (received all desired services) | 1801 (68.6) |
| Did not receive 1 desired service | 604 (23.0) |
| Did not receive 2 desired services | 166 (6.3) |
| Did not receive 3 or more desired services (up to 8) | 56 (2.1) |
| **Types of services wanted during pregnancy but not received[a] (n = 826)** | |
| Pregnancy/childbirth classes | 498 (60.3) |
| Food banks | 186 (22.5) |
| Ultrasound scans, birth defect screenings (e.g., trisomy 21), medical tests (e.g., blood test, cervical exams, physicals) or appointments with HCP | 141 (17.1) |
| Family services (e.g., child care, counselling, parenting courses) or support services (e.g., mental health services) | 124 (15.0) |
| Other services (including housing assistance and traditional medicines) | 184 (22.3) |
| **Number of barriers experienced precluding access to desired pregnancy care (n = 2616)** | |
| No barriers | 2018 (77.1) |
| 1 barrier | 459 (17.5) |
| >1 barrier | 139 (5.3) |
| **Barrier(s) experienced precluding access to desired pregnancy care[a] (n = 598)** | |
| Did not realize services were offered; did not know where they were offered; or mother did not realize she was eligible for these services | 393 (65.7) |
| Did not have the time; had to work; or forgot to seek/attend appointment | 234 (39.1) |
| Services were not offered in the area; services were already full; HCP did not follow up on request for care; or HCP cancelled the appointment | 181 (30.3) |
| Was not eligible for these services; administrative barriers (e.g., no insurance, late registration); or thought it was too late for care | 78 (13.0) |
| Needed to stay at home; childcare was not available; or did not have access to transportation | 65 (10.9) |
| Had difficulties understanding how the healthcare system works and/or problems using the healthcare system | 52 (8.7) |
| Language barrier | 39 (6.5) |
| The care mother received was not what she was expecting from the healthcare system | 29 (4.8) |
| Financial reasons | 21 (3.5) |
| Other reason | 17 (2.8) |

[a]Participants could choose all options that applied, therefore the percentages total >100%.

**Table 4. MFMC components provided: Communication, all phases of maternity care.**

| *Communication during pregnancy* | n (%) |
|---|---|
| **HCP gave information about pregnancy in mother's own language (n = 2633)** | 1141 (43.3) |
| **Mother felt she received enough information during pregnancy (n = 2633)** | |
| Enough information on all topics asked | 917 (34.8) |
| Not enough information on >3 topics | 651 (24.7) |
| Not enough information on 2 or 3 topics | 605 (23.0) |
| Not enough information on one topic | 460 (17.5) |
| **Sources of information during pregnancy (n = 2631)[a]** | |
| Internet, including cellphone applications | 1618 (61.5) |
| Books, including magazines and pamphlets | 1070 (40.7) |
| Previous pregnancy | 1025 (39.0) |
| Family/Friends | 938 (35.7) |
| Obstetrician-gynecologist (OB/GYN) | 757 (28.8) |
| Pregnancy or childbirth classes | 222 (8.4) |
| Nurse, including *CLSC* (community health service centre) or *Info-Santé* (health phone service) | 191 (7.3) |
| Spiritual or religious sources | 141 (5.4) |
| *Communication during labour* | n (%) |
| **HCP gave information about breastfeeding when needed (n = 2013)** | 1402 (69.6) |
| **HCP offered mother help with breastfeeding (n = 2606)** | |
| In the first hour after birth | 1642 (63.0) |
| Not immediately, but still in location of birth (i.e., hospital) | 877 (33.7) |
| At a later appointment, or not at all | 87 (3.3) |
| *Communication across three phases[b] or overall[c]* | n (%) |
| **Mother *always* felt HCP kept her informed (n = 2621)[b]** | 2257 (86.1) |
| **Mother *always* understood information provided by HCP (n = 2626)[b]** | 2283 (86.9) |
| **Mother would have understood information provided by HCP *better* in another language (n = 2634)[c]** | 757 (28.7) |
| **HCP did *not* offer interpreting service (n = 847)[b,d]** | 717 (84.7) |
| **Someone who could interpret was *always* in attendance for women in need (n = 742)[b,d]** | 388 (52.3) |
| **There was *never* someone who could interpret in attendance for women in need (n = 742)[b,d]** | 161 (21.7) |
| ***HCP or professional* interpreted at least once (n = 736)[b,d]** | 25 (3.4) |
| ***Someone else* (e.g., family, friend) interpreted at least once (n = 736)[b,d]** | 454 (61.7) |
| ***No one* ever interpreted in mother's language (n = 736)[b,d]** | 153 (20.8) |
| **Mother was satisfied with interpretation when needed (n = 564)[d]** | 532 (94.3) |

[a]Participants could chose all options that applied, therefore the percentages total >100%.

[b]These questions were phase dependent, meaning participants answered questions for the pregnancy, labour and post-birth phases; results were combined because variation was <10% across phases. Responses presented here show the number of participants who responded the same for all three phases.

[c]These questions were not phase dependent.

[d]Number of women for whom the question was applicable.

Few women reported being asked about preferences for care during pregnancy (8.7%), labour (11%) or after birth (4.2%), and most (92.6%) reported never being asked about their preferred gender for HCP. However, the majority (84.4%) of women were asked about pain management during labour and 87.6% were satisfied with how their pain was managed. Almost all women (99.1%) had their choice of support people during labour, and most mothers (92.4%) felt that

**Table 5. MFMC components provided: Experiences & perceptions of care, all phases of maternity care.**

| *Care experiences during pregnancy* | n (%) |
|---|---|
| **Healthcare provider (HCP) asked about preferences for care during pregnancy (n=2629)** | 229 (8.7) |
| **HCP asked how woman planned to feed infant during pregnancy (n=2618)** | 2254 (86.1) |
| *Care experiences during labour* | n (%) |
| **HCP asked about preferences for care during labour (n=2613)** | 287 (11) |
| **HCP asked woman about pain management during labour (n=2165)** | 1828 (84.4) |
| **Satisfaction with pain management during labour (n=2127)** | |
| Satisfied | 1864 (87.6) |
| Sometimes satisfied, or not satisfied at all | 263 (12.4) |
| **Woman allowed to move around and choose positions during labour (n=1969)** | |
| Always | 1417 (72.0) |
| Sometimes | 456 (23.2) |
| Rarely or not at all | 96 (4.9) |
| **Woman had her choice of support people during labour (n=2212)** | 2191 (99.1) |
| **Mother *always* had a companion during birth (n=2619)** | 2466 (94.2) |
| *Care experiences post-birth* | n (%) |
| **HCP asked about preferences for care after birth (n=2634)** | 110 (4.2) |
| **Mother was given baby to hold skin-to-skin in first hour post-birth (n=2634)** | 2406 (91.3) |

**Care experiences within three measured phases, n (%)**

| HCPs could do differently/better | *Pregnancy (n=2599)* | *Labour (n=2602)* | *Post-birth (n=2594)* |
|---|---|---|---|
| Yes | 736 (28.3) | 388 (14.9) | 581 (22.4) |
| No | 1863 (71.7) | 2214 (85.1) | 2013 (77.6) |
| **Mother waited too long for care** | *Pregnancy (n=2628)* | *Labour (n=2628)* | *Post-birth (n=2629)* |
| Always | 330 (12.6) | 97 (3.7) | 80 (3.0) |
| Sometimes | 693 (26.4) | 219 (8.3) | 267 (10.2) |
| Rarely | 263 (10.0) | 183 (7.0) | 270 (10.3) |
| Never | 1342 (51.1) | 2129 (81.0) | 2012 (76.5) |

| *Care experiences across three phases[a] or overall[b]* | n (%) |
|---|---|
| **Mother *never* felt discriminated against by HCP (n=2628)** | 2487 (94.6) |
| **Perceived reasons for discrimination, if applicable (n=130)[c]** | |
| Language | 48 (36.9) |
| Race | 34 (26.2) |
| Culture | 25 (19.2) |
| Religion | 20 (15.4) |
| Unsure why (write-in response) | 19 (14.6) |
| Colour | 13 (10.0) |
| Migration status, insurance status, marital status, or other | 25 (19.2) |
| **Mother *always* felt respected by HCP (n=2616)[a]** | 2473 (94.5) |
| **HCPs *never* made decisions without considering woman's wishes (n=2613)[a]** | 2415 (92.4) |
| **Mother *always* felt comfortable asking about things not understood (n=2626)[a]** | 2377 (90.5) |
| **Mother *always* found HCPs helpful (n=2620)[a]** | 2367 (90.3) |
| **Mother *always* felt welcomed by HCP (n=2612)[a]** | 2304 (88.2) |
| **Mother *always* felt concerns were taken seriously by HCP (n=2613)[a]** | 2208 (84.5) |
| **HCPs were *always* encouraging and reassuring (n=2622)[a]** | 2208 (84.2) |
| **Mother was *always* happy with care received (n=2624)[a]** | 2189 (83.4) |
| **HCPs *always* spent enough time providing explanations (n=2621)[a]** | 2082 (79.4) |
| **HCP asked if mother had questions (n=2635)[b]** | |

*(Continued)*

**Table 5.** (Continued)

| Care experiences during pregnancy | n (%) |
|---|---|
| Always | 2085 (79.1) |
| Sometimes | 394 (15.0) |
| Rarely or never | 156 (5.9) |
| **HCPs were rushed (n = 2635)[b]** | |
| Always | 209 (7.9) |
| Sometimes or rarely | 905 (34.3) |
| Never | 1521 (57.7) |
| **HCP asked about mother's preferences for food (n = 2626)[b]** | 1284 (48.9) |
| **Mothers were *never* asked about preferences for male/female HCP (n = 2622)[a]** | 2427 (92.6) |

[a]These questions were phase dependent, so participants answered questions for the pregnancy, labour and post-birth phases; results were combined because variation was < 10% across phases. Responses presented here show the number of participants who responded the same for all three phases.

[b]These questions were not phase dependent.

[c]Participants chose all options that applied, therefore the percentages total >100%. Although 141 women reported feeling discriminated, only 130 responded to the question regarding reasons why.

HCPs did not make decisions without taking their wishes into account, regardless of the phase of care (pregnancy, childbirth or postpartum).

### 3.3. Relationship between sociodemographic and migration background and extent of MFMC provided

Results of logistic regression analyses are reported in Table 6.

**3.3.1. MFMC during pregnancy.** Access to prenatal care was reported as 35% better for mothers with medium French/English language ability compared to those with high French/English language ability (OR 1.35, 95% CI 1.01–1.81) and reported as twice as good for mothers from East/South-East Asia compared to those who had migrated from other regions (2.02, 1.28–3.20). Prenatal care access was reported as 45% worse for migrants in Canada <2 years (0.55, 0.36–0.85), and 50% worse for migrants 2−5 years in Canada (0.50, 0.31–0.80) compared to migrants in Canada >5 years. It was also reported as 31% worse for mothers with lower household incomes compared to mothers with higher incomes (0.69, 0.52–0.90), and 34% worse for mothers with less education compared to those with more education (0.66, 0.47–0.94).

The odds of reporting better communication facilitation during pregnancy were 51% higher for mothers who had never given birth in Canada before (1.51, 1.19–1.92), 44% higher for low-income mothers (1.44, 1.19–1.76), and over twice as high for mothers from Sub-Saharan Africa (2.24, 1.76–2.85). Communication facilitation during pregnancy was reported as 39% worse for primiparous mothers (0.61, 0.49–0.77), 25% worse for those with medium French/English language ability (0.75, 0.61–0.91), 79% worse for those with low French/English language ability (0.21, 0.14–0.32), and 32% worse for those from Eastern Europe (0.68, 0.47–0.98).

HCP support during pregnancy was reported as 50% better for migrants having been in Canada for 2−5 years (1.50, 1.01–2.22), while it was reported as 44% worse for those with low French/English language ability (0.56, 0.36–0.87) and 33% worse for those from the Middle-East/North Africa (0.77, 0.59–0.99).

Responsiveness to preferences for care during pregnancy was reported as 36% worse for mothers from the Middle-East/North Africa (0.64, 0.47–0.89).

**3.3.2. MFMC during labour.** The odds of reporting better communication facilitation during labour and birth were over three times as high for mothers from Sub-Saharan Africa (OR 3.24, 95% CI 1.81–5.80), while they were 43% lower for

**Table 6. Explanatory variables associated with Migrant-Friendly Maternity Care (MFMC)-related outcomes: OR (95% CI)¹ [N=2636]².**

| | | MFMC outcomes during pregnancy | | | | MFMC outcomes during labour | | |
|---|---|---|---|---|---|---|---|---|
| Socioeconomic characteristics & migration descriptors | | Access to prenatal care (N=2163) | Communication facilitation (N=2208) | HCP support (N=2202) | Responsiveness to preferences (N=2202) | Communication facilitation (N=2207) | HCP support (N=2191) | Responsiveness to preferences (N=2194) |
| Previous birth in Canada | No | | 1.51 (1.19, 1.92) | | | | | 1.30 (0.90, 1.90) |
| | Yes | | 1.00 | | | | | 1.00 |
| Time in Canada | <2 years | 0.55 (0.36, 0.85) | | 1.24 (0.88, 1.75) | | | | |
| | 2-5 years | 0.50 (0.31, 0.80) | | 1.50 (1.01, 2.22) | | | | |
| | >5 years | 1.00 | | 1.00 | | | | |
| Parity | Primiparous | | 0.61 (0.49, 0.77) | | | | 1.38 (1.06, 1.80) | 1.12 (0.80, 1.56) |
| | Multiparous | | 1.00 | | | | 1.00 | 1.00 |
| French/English spoken ability | Low | 1.01 (0.62, 1.64) | 0.21 (0.14, 0.32) | 0.56 (0.36, 0.87) | | 0.08 (0.06, 0.13) | | 0.55 (0.31, 0.98) |
| | Medium | 1.35 (1.01, 1.81) | 0.75 (0.61, 0.91) | 0.90 (0.67, 1.20) | | 0.57 (0.42, 0.78) | | 0.82 (0.61, 1.10) |
| | High | 1.00 | 1.00 | 1.00 | | 1.00 | | 1.00 |
| Income | Lower income | 0.69 (0.52, 0.90) | 1.44 (1.19, 1.76) | | | | 0.72 (0.55, 0.94) | |
| | Higher income | 1.00 | 1.00 | | | | 1.00 | |
| Education | ≤Sec. education | 0.66 (0.47, 0.94) | | | | | | |
| | >Sec. education | 1.00 | | | | | | |
| Maternal age | ≥35 years old | | | | | | | |
| | <35 years old | | | | | | | |
| Region of birth | Sub-Sah. Africa | 2.02 (1.28, 3.20) | 2.24 (1.76, 2.85) | | 0.66 (0.42, 1.03) | 3.24 (1.81, 5.80) | 0.74 (0.44, 1.25) | |
| | M. East/N. Africa | | | 0.77 (0.59, 0.99) | 0.64 (0.47, 0.89) | 1.20 (0.81, 1.79) | 0.65 (0.41, 1.03) | 0.71 (0.52, 0.96) |
| | East/S.-East Asia | | | | | 0.45 (0.30, 0.69) | 1.32 (0.73, 2.38) | 0.87 (0.57, 1.31) |
| | South Asia | | | | | 0.80 (0.48, 1.34) | 0.95 (0.50, 1.81) | |
| | Eastern Europe | | 0.68 (0.47, 0.98) | | | 0.80 (0.47, 1.37) | 0.64 (0.35, 1.18) | 0.84 (0.49, 1.43) |
| | South America | | | | | | | |
| | Other regions³ | 1.00 | 1.00 | 1.00 | 1.00 | 1.00 | 1.00 | 1.00 |
| Constant | | 11.11 | 0.60 | 7.12 | 0.12 | 8.92 | 8.87 | 0.14 |

¹ORs > 1 convey a beneficial association. ORs <1 convey an adverse association. Empty cells indicate variables that were removed from the model.

²In each of the 7 models, only women with data on all potential explanatory variables and outcomes were included. Note that a total of 2222 women answered the question on income, thus excluding 414 women from the total sample for inclusion in the models.

³A separate regression model was generated for each region. The reference category for each region was every participant *not* from that region.

women with medium French/English language ability (OR 0.57, 95% CI 0.42–0.78), 92% lower for those with low French/English language ability (OR 0.08, 95% CI 0.06–0.13), and 55% lower for those from East/South-East Asia (OR 0.45, 95% CI 0.30–0.69).

HCP support during labour and birth was reported as 38% better for primiparous mothers (OR 1.38, 95% CI 1.06–1.80), and 28% worse for mothers with low household incomes (OR 0.72, 95% CI 0.55–0.94).

Responsiveness to preferences for care during labour and birth was reported as 45% worse for mothers with low French/English language ability (OR 0.55, 95% CI 0.31–0.98), and 29% worse for women who migrated from the Middle-East/North Africa (OR 0.71, 95% CI 0.52–0.96).

## 4. Discussion

### 4.1. Extent of MFMC provided

The results of this study suggest that many recommended components of MFMC have been implemented in Montreal, though it appears that these are better provided during labour than during pregnancy. For example, almost all participants were accompanied, never waited too long for care, and had effective pain management during labour, while access to services and information, waiting too long for care, and timely prenatal care (for some subgroups) remained a challenge during pregnancy. The most common access barrier experienced during pregnancy was a lack of information in terms of availability and eligibility criteria for services. This aligns with prior findings showing that migrant women find it difficult to navigate the health and social care system, including knowing what services are available and their eligibility requirements. [30,52–54]

Across all phases of maternity care, most women reported that HCPs were respectful, welcoming, and always kept them informed, and that they always understood the information provided. Most women also reported feeling reassured and encouraged by HCPs. However, participants were rarely asked about their preferences for care, including their preferred gender of HCP. Other studies have reported women's preferences not being honoured, and show that when asked, migrant women generally prefer female HCPs. [19,55]

Regarding communication facilitation, most women who needed interpreter services reported not being offered any, though almost all were satisfied with such services when received. A significant proportion of women also reported that a family member or friend provided interpretation at least once, though this approach goes against recommendations. [56,57] The unmet need for translators among migrants has been identified elsewhere in the literature and has often been an intervention of focus as a modifiable aspect of care. [52,56]

### 4.2. Relationship between sociodemographic and migration background and extent of MFMC provided

Our results suggest that the participants' socio-demographic and migration characteristics influence their care experiences during pregnancy and labour. Indeed, all the explanatory variables measured, except for maternal age, were associated with one or more MFMC components (access to prenatal care, communication facilitation, support by HCPs and responsiveness to preferences for care).

The variables with the most notable influence on MFMC outcomes in pregnancy and labour were French/English language ability and region of maternal birth. After controlling for other characteristics, French/English language ability was most consistently associated with receiving inadequate MFMC, except for access to prenatal care, while region of maternal birth was associated, both positively and negatively, with all care outcomes, except for HCP support during labour.

Our findings on the effect of French/English language ability, with the exception of access to prenatal care, [25] align with prior studies showing a negative relationship between care and low host-country language ability. [7,8,19,27,44] Addressing the challenge of communication requires, at minimum, HCPs being able to effectively communicate with patients in their language or through appropriate interpreter services or other translation tools. As international migration continues to increase, so will language diversity, and addressing communication is key to establishing trust and a positive

maternity care experience. [46,52] Furthermore, fluency in the host-country language doesn't necessarily mean that medical terms will be understood, so interpretation services can also benefit those who may speak the language but prefer to have information given in a language they are more comfortable with. [58]

Regarding maternal region of birth, studies have shown varying experiences with maternity care between migrant women from various regions of birth and non-migrants. [30] Our study further highlights the effect of different regions of maternal birth among migrant groups on perceptions and experiences with care. These findings may reflect differences in patient behaviours and expectations for care, but also differences in care delivery. [25,30,47,52,59–61] Taking time to provide women information about the process of giving birth in the host country context may help them better predict what to expect, and in turn improve experiences of care. Further, maintaining cultural sensitivity and providing responsive care – including asking patients about their preferences, ensuring continuity of care, and improving communication between HCPs and patients – would improve patient experiences. [52,55]

Finally, our findings reflect that the most vulnerable migrants – the more recently-arrived and those with lower income and less education – experienced worse access to prenatal care, which aligns with the literature. [32,48] These findings, taken together with the types of barriers reported precluding access to services, are suggestive of a larger structural challenge with information dissemination. Accessing prenatal care on time requires effective outreach to those that need it. Since 2022, to improve access to maternity care for asylum seekers, PRAIDA- the Regional program for the settlement and integration of Asylum Seekers in Québec, implemented a triage system in which pregnant women are screened and directly referred to a hospital for follow-up care. However, no specific policies to improve access for other migrant groups, have been introduced. This lack has partially been addressed by making pregnancy care information more available through migrant community organizations and government websites – sources that pregnant migrants usually access to become informed about maternity topics. Most participants reported using the internet or phone applications, so this approach could be further leveraged to improve access to information about the availability of services and the eligibility criteria for different migrant groups. Other solutions include improving patient navigation of the healthcare system with the help of dedicated individuals to help orient new patients. Despite experiencing poorer access to prenatal care, these vulnerable migrants appeared to receive better communication support once they accessed services, suggesting that, once they are within the system, they are being directed toward resources. This observation highlights the essential fact that improving MFMC to these groups should begin with bettering their access to care.

When considering the effect of sociodemographic and migration characteristics, it is important to note that the sample was very highly educated (98.1% had at least a secondary education), such that the at-risk subgroup was defined as ≤ secondary education completed, which is not comparable to how low-education is usually defined in the literature. [7,25] Our sample, however, is representative of migrant women from non-Western countries living in Quebec at the time of the study, with relatively high proportions of well-educated, French-speaking migrants, and low numbers of refugees. This reflects Quebec's language and education requirements for immigration to the province, [62] and Canada's immigration policy at the time, which favoured economic migrants. [63]

In this study, we measured different MFMC components, including several different sub-elements, experienced by over 2,500 international migrant women giving birth in an urban centre. Our results show that to optimize MFMC, delivery of services should account for the various and disparate expectations, experiences, and characteristics of each woman and her family. Improving information dissemination, providing access to interpreter services, and adapting the way care is given at the point of care – for example spending more time with vulnerable patients and asking about preferences for care – are all essential recommendations supported by our findings. We hope that inquiry on migrant women's maternity care experiences will continue, not only to expand on the findings presented here, but also to provide updated insights into the evolving realities of their care. In particular, the study of specific subgroups such as humanitarian migrants, which we were not able to examine due to small numbers, could offer useful insights. More in-depth study, to better understand the mix of associations found between different regions of maternal birth and various care outcomes, would also be informative.

### 4.3. Strengths & limitations

This study had several strengths including: research methods that were developed in collaboration with migrant community members; representativeness of the international migrant population of Montreal at the time of the study; use of the MFMCQ, a validated tool that has been used internationally; multi-lingual interviewers; questionnaires available in multiple languages; and a focus on women's views of different components of care, which are potentially amenable to change through policy and practice. There was a rigorous recruitment process, which included screening more than 11,000 births in four urban hospitals for study eligibility, ultimately leading to data from 2,636 migrant women, from which over 2,100 were included for regression analyses. The MFMCQ captured different components of MFMC and included several different sub-elements, thus allowing for nuanced, detailed analyses of women's experiences and perceptions of care during pregnancy, labour and postpartum.

Our study also had weaknesses. We excluded some women due their vulnerability (e.g., those who had a stillbirth), we were not able to determine the eligibility of all women who gave birth during the study period, and we removed participants with missing information from our regression analyses, so it is possible that the results are not fully representative. Although the MFMCQ was tested and validated, some questions still lacked clarity. We therefore chose to rely on questions that were clearly understood, and, when drawing conclusions, we considered the results from a combination of questions to optimize validity. One of the MFMC component measures, HCP support during pregnancy, was defined based on a single question (were HCPs reassuring and/or encouraging). Other questions that were related to HCP support during pregnancy were deemed better placed under different MFMC components. Finally, the low variability in responses for certain migration and sociodemographic variables prohibited us from conducting more refined analyses.

High levels of reported satisfaction with care could be attributed in part to a halo effect, since our data were collected on the post-partum units of hospitals very soon after birth. This effect has been described elsewhere where mothers highly regard their experiences of care immediately after birth but their positive impressions weaken with time. [53,55] This same bias has also been seen in recently-arrived migrants experiencing healthcare in a Western country for the first time; given the high number of participants who were giving birth for the first-time in Canada (67.1%), it is plausible that this bias was present in our study. [53,55] Social desirability could also play a role in the high satisfaction reported, as the women were interviewed in hospital and could have been reluctant to report negative care experiences while still in the post-partum unit. [53] Due to the cross-sectional design, we cannot explain the relationships between the socioeconomic and migration variables and the MFMC outcomes, and the results only reflect one point in time. Moreover, since the data were collected 10 years ago, it's possible that the population and migrants' maternity experiences have changed and the results do not capture the current situation. However, there has been limited policy change in recent years regarding migrants and maternity healthcare in Québec, so the issues related to MFMC identified in this study are likely still relevant. Furthermore, the healthcare system has faced increasing strain, largely due to the pandemic, and migrants' vulnerabilities have intensified as a result of ongoing sociopolitical and economic challenges, suggesting that care issues may have worsened over time.

Taken together, the study strengths were many, and the weaknesses were few, with the impact of the latter on our conclusions largely mitigated. Overall, our study gathered a wealth of data on various aspects of MFMC on over 2,500 women giving birth in an urban city, and provides for specific, nuanced recommendations for both policy and care.

### 4.4. Conclusions

Overall, participants in this study felt care was helpful and respectful, and they found HCPs encouraging and reassuring. However, navigating care during pregnancy was a challenge and resulted in some women not receiving the care or services they wanted. Access to prenatal care was worse for certain subgroups, including more recently-arrived migrants, and those with less education and lower incomes. Additionally, language barriers presented meaningful challenges throughout the continuum of maternity care, and interpretation services were lacking despite a demonstrable need. HCPs

also rarely asked about preferences for care. Furthermore, our findings showing differing effects of region of maternal birth on MFMC components suggest that past experiences and/or cultural differences may shape expectations and influence how care in a new country is perceived and experienced. It may also indicate real differences in the delivery of care to certain groups.

Our results show that further improvements to MFMC are warranted, particularly for women with vulnerable sociodemographic and migration backgrounds. Notably, HCPs and policymakers should consider and account for host-country language ability, length of time in the country, preferences for care and the need for support in navigating a new healthcare system. The successful implementation of MFMC depends on a number of contextual factors, including available resources, HCPs' perceptions of migrants, documentation and communication of migration information, culturally competent maternity care training, recruitment and retention of culturally and linguistically diverse healthcare personnel, and protocols adapted to consider migrant women's care needs. [34,37] Harnessing these modifiable contextual factors to improve navigation, interpretation and expectations for care would prove most useful in bettering MFMC.

## Supporting information

**S1 Appendix. MFMCQ Translation Protocol.**
(PDF)

**S2 Appendix. MFMCQ Arabic Version.**
(PDF)

**S3 Appendix. MFMCQ English Version.**
(PDF)

**S4 Appendix. MFMCQ French Version.**
(PDF)

**S5 Appendix. MFMCQ Hindi Version.**
(PDF)

**S6 Appendix. MFMCQ Chinese (Complex Characters) Version.**
(PDF)

**S7 Appendix. MFMCQ Spanish Version.**
(PDF)

**S8 Appendix. MFMCQ Urdu Version.**
(PDF)

**S9 Appendix. MFMCQ Vietnamese Version.**
(PDF)

**S10 Appendix. MFMCQ Supplemental Questionnaire – Arabic Version.**
(PDF)

**S11 Appendix. MFMCQ Supplemental Questionnaire – English Version.**
(PDF)

**S12 Appendix. MFMCQ Supplemental Questionnaire – French Version.**
(PDF)

**S13 Appendix. MFMCQ Supplemental Questionnaire – Hindi Version.**
(PDF)

**S14 Appendix. MFMCQ Supplemental Questionnaire – Chinese (Complex Characters) Version.**
(PDF)

**S15 Appendix. MFMCQ Supplemental Questionnaire – Spanish Version.**
(PDF)

**S16 Appendix. MFMCQ Supplemental Questionnaire – Urdu Version.**
(PDF)

**S17 Appendix. MFMCQ Supplemental Questionnaire – Vietnamese Version.**
(PDF)

**S18 Appendix. Medical Record Review Form.**
(PDF)

## Acknowledgments

The authors would like to acknowledge the invaluable contributions of Theresa W. Gyorkos, Masaud Kakkar, Johane Lorvinsky, Jessica Seferian, the full research team, the ROAM collaboration, and the collaborating hospitals, as well as the participation of 2,636 migrant women.

## Author contributions

**Conceptualization:** Lisa Merry, William Fraser, Sonia Semenic, Sandra Pelaez, Ayesha Baig, Anita Gagnon.

**Data curation:** Lisa Merry, Ayesha Baig.

**Formal analysis:** Isabel Baltzan, Lisa Merry, Ayesha Baig.

**Funding acquisition:** Lisa Merry, William Fraser, Sonia Semenic, Anita Gagnon.

**Investigation:** Alexis Edington, Ayesha Baig.

**Methodology:** Lisa Merry, William Fraser, Sonia Semenic, Anita Gagnon.

**Project administration:** Alexis Edington, Ayesha Baig.

**Resources:** Ayesha Baig, Anita Gagnon.

**Supervision:** Ayesha Baig, Anita Gagnon.

**Validation:** Lisa Merry, Anita Gagnon.

**Visualization:** Isabel Baltzan, Lisa Merry.

**Writing – original draft:** Isabel Baltzan, Anita Gagnon.

**Writing – review & editing:** Lisa Merry, William Fraser, Sonia Semenic, Sandra Pelaez, Alexis Edington, Ayesha Baig, Anita Gagnon.

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
