## [Decision Letter · Decision Letter 0]

18 May 2025

Dear Dr. Merry,

Thank you for submitting your manuscript to PLOS ONE. After careful consideration, we feel that it has merit but does not fully meet PLOS ONE’s publication criteria as it currently stands. Therefore, we invite you to submit a revised version of the manuscript that addresses the points raised during the review process.

We look forward to receiving your revised manuscript.

Kind regards,

Rebecca F. Baggaley

Academic Editor

PLOS ONE

Journal Requirements:

“Funding for this research was provided by the Canadian Institutes of Health Research (CIHR).”

Reviewers' comments:

Reviewer's Responses to Questions

**Comments to the Author**

1. Is the manuscript technically sound, and do the data support the conclusions?

Reviewer #1: Yes

Reviewer #2: Yes

2. Has the statistical analysis been performed appropriately and rigorously?

Reviewer #1: Yes

Reviewer #2: Yes

3. Have the authors made all data underlying the findings in their manuscript fully available?

Reviewer #1: Yes

Reviewer #2: Yes

4. Is the manuscript presented in an intelligible fashion and written in standard English?

Reviewer #1: Yes

Reviewer #2: Yes

Reviewer #1: This is a perfectly publishable study. The specific restrictions on data availability are justified in the manuscript. The strengths and limitations section is considered and fair. I would question how the service landscape/quality may have changed since data collection as this was 10 years ago. Other than that I have no hesitation in recommending this article for publication.

Reviewer #2: Although the study is well designed, described and discussed after the findings, there are a few places where adding some explanations or wording could improve the understanding for the readers. Please see them in the attached file. Thanks.

**Do you want your identity to be public for this peer review?** For information about this choice, including consent withdrawal, please see our Privacy Policy

Reviewer #1: No

Reviewer #2: **Yes: ** Dr. Alfredo L. Fort

---

## [Author Response · Author response to Decision Letter 1]

17 Jul 2025

Please see the attached file entitled "Response to reviewers"

---

## [Editor Report · Decision Letter 1]

7 Aug 2025

Migrant-friendly maternity care in Montreal, Canada: A cross-sectional study on migrant women’s care perspectives

PONE-D-24-49803R1

Dear Dr. Merry,

We’re pleased to inform you that your manuscript has been judged scientifically suitable for publication and will be formally accepted for publication once it meets all outstanding technical requirements.

Kind regards,

Rebecca F. Baggaley

Academic Editor

PLOS ONE
---

## [Editor Report · Acceptance letter]

PONE-D-24-49803R1

PLOS ONE

Dear Dr. Merry,

I'm pleased to inform you that your manuscript has been deemed suitable for publication in PLOS ONE. Congratulations! Your manuscript is now being handed over to our production team.

Kind regards,

on behalf of

Dr. Rebecca F. Baggaley

Academic Editor

PLOS ONE